# Sleep Quality, Anxiety, and Depression Are Associated with Fall Risk Factors in Older Women

**DOI:** 10.3390/ijerph17114043

**Published:** 2020-06-05

**Authors:** Rodrigo Serrano-Checa, Fidel Hita-Contreras, José Daniel Jiménez-García, Alexander Achalandabaso-Ochoa, Agustín Aibar-Almazán, Antonio Martínez-Amat

**Affiliations:** 1Department of Health Sciences, Faculty of Health Sciences, University of Jaén, 23071 Jaén, Spain; rserranocheca@gmail.com (R.S.-C.); fhita@ujaen.es (F.H.-C.); aaochoa@ujaen.es (A.A.-O.); aaibar@ujaen.es (A.A.-A.); amamat@ujaen.es (A.M.-A.); 2MOVE-IT Research Group and Department of Physical Education, Faculty of Education Sciences, University of Cádiz, 11003 Cádiz, Spain; 3Biomedical Research and Innovation Institute of Cádiz (INiBICA) Research Unit, Puerta del Mar University Hospital, University of Cádiz, 11009 Cádiz, Spain

**Keywords:** sleep quality, anxiety, depression, gait speed, functionality, dynamic balance

## Abstract

Gait, dynamic balance, and functional mobility problems are well-known fall risk factors. Furthermore, sleep disturbances, anxiety, and depression are prevalent among older women. This study aimed to analyze the associations of sleep quality, anxiety, and depression with functional mobility, gait speed, and dynamic balance in community-dwelling postmenopausal women aged ≥ 60 years. A total of 271 women (69.18 ± 5.69 years) participated in this study. Functional mobility (Timed Up-and-Go Test), dynamic balance (3-meter tandem walk test), gait speed (OptoGait® optical detection system), sleep quality (Pittsburgh Sleep Quality Index), and anxiety and depression (Hospital Anxiety and Depression Scale) were assessed. Our results showed that poor sleep efficiency and the use of sleeping medication were related to decreased gait speed (R^2^ = 0.072). Poor functional mobility was linked to depression and the use of sleeping medication (R^2^ = 0.159). Additionally, increased symptoms of anxiety and depression were associated with worsened dynamic balance (R^2^ = 0.127). In conclusion, poorer sleep quality is associated with slower gait speed and reduced functional mobility, which is also related, along with impaired dynamic balance, to higher levels of anxiety and depression.

## 1. Introduction

Sleep disturbances, anxiety, and depression are prevalent among postmenopausal women [1,2], who in addition are more commonly affected by sleep difficulties than younger women [3]. Furthermore, during the post-menopausal stage between 40% and 60% of women are affected by poor sleep and insomnia [4]. Poor sleep quality and quality of life are associated with physical and psychological problems [5,6].

As far as psychological problems are concerned, anxiety and depression are very common and debilitating diseases with a considerable impact on the economic, social, and personal life of those affected [7,8]. Women are at higher risk of depressive disorder than men [9]. This is usually related to certain stressors (i.e., familial, economic, social, sexual, and health-related) that seem to aggravate mood disorders in this population [10,11,12,13]. Menopausal status is also associated with an increased risk of mood disorders. Among women without a history of depression or anxiety, and compared with pre-menopausal individuals, the peri-and postmenopausal stages are associated with an increased risk for symptoms of anxiety [14].

Aging is generally associated with gait and balance disorders, neuromuscular impairment, and lower levels of mobility, and therefore with an increased risk of falls and fractures [15]. Falls and fall-related injuries are one of the main health problems affecting postmenopausal women [16]. Due to the aging process, elderly and postmenopausal individuals tend to reduce their physical activity levels and become more dependent on external help. Falls are the result of several multivariate risk factors which are frequently classified as either extrinsic or intrinsic. The later are related to functional and health status, with gait instability and balance deficit being some of the most common intrinsic risk factors in older people [17]. Gait speed has repeatedly been shown to be a reliable tool in predicting falls and discriminating between fallers and non-fallers [18]. Decreased gait speed [19], poor mobility, and dynamic balance [20] have been linked to an increased risk of falling among community-dwelling older adults. 

This study aims to analyze the associations of sleep quality, anxiety, and depression with functional mobility, gait speed, and dynamic balance in community-dwelling postmenopausal women aged 60 years and older. We hypothesize that postmenopausal individuals displaying signs of better sleep quality and lower levels of anxiety and depression have faster gait speed as well as better functional mobility and dynamic balance.

## 2. Materials and Methods 

### 2.1. Study Participants

This cross-sectional study involved community-dwelling women who were recruited through several associations of postmenopausal women in the Eastern Andalusia region (Jaén and Málaga), and using municipal records, local media, and social networks. A total of 271 older women (69.18 ± 5.69 years) participated in this study (Figure 1). To be included, subjects had to be women aged 60 years and over and able to understand the instructions, programs, and protocols involved in the study. Exclusion criteria were conditions that contraindicated the performance of physical tests, diseases that could alter balance and functional activity (such as auditory or vestibular alterations), central or peripheral neurological disorders, and serious psychiatric or somatic diseases. Firstly, we informed subjects about the main purpose and the reasons for conducting the study. Secondly, we briefly explained its associated risks. All procedures were anonymous. A written informed consent form was obtained from each participant before enrollment. This study was approved by the local Human Ethics Committee (OCT.18/4.PRY) and was conducted in accordance with the Declaration of Helsinki, good clinical practices, and applicable laws and regulations.

### 2.2. Outcomes

Outcomes measures were sociodemographic and anthropometric data, sleep quality, anxiety and depression, functional mobility, dynamic balance and gait speed.

#### 2.2.1. Sociodemographic and Anthropometric Data 

Demographic data such as age, years of menopause, educational, marital, and occupational status, and smoking habits were collected by well-trained interviewers. Body mass index (BMI) was obtained by dividing the women’s weight (kg) by their height squared (m^2^). A 100 g–130 kg precision digital weight scale (Tefal, Barcelona, Spain) and T201-T4 Asimed adult height scale (Asimed, T201-T4, Barcelona, Spain) were employed for weight and height assessment. A BMI ≥ 30 kg/m^2^ indicated obesity [21]. Waist circumference was measured using a 1.5 m flexible tape (Lufkin, W606PM, MD, USA) at the midpoint between the lowest rib and the iliac crest, with participants in the standing position. A waist circumference ≥88 cm was taken as an indicator of abdominal obesity [21].

#### 2.2.2. Sleep Quality

In order to assess sleep quality, the Pittsburgh Sleep Quality Index (PSQI) [22,23] was used. This questionnaire includes 19 self-rated questions and 5 questions to be answered by bedmates or roommates (these last only being used to gather clinical information). The 19 self-rated questions generate a total score and 7 domain scores (on a scale of 0 to 3, with higher scores indicating poorer sleep quality): C1, sleep quality; C2, sleep latency; C3, sleep duration; C4, sleep efficiency; C5, sleep disturbances; C6, use of sleep medication; C7, daytime dysfunction. In turn, these 7 scores range from 0 to 21, with a total score >5 indicating poor sleep quality [22,24].

#### 2.2.3. Anxiety and Depression

The Hospital Anxiety and Depression Scale (HADS) [25,26] was developed to assess anxiety and depression in the general population. A total of 14 items are equally divided into an anxiety subscale of 7 items and a depression subscale of the remaining other 7 items [27]. Scores range from 0–21, where higher values indicate more severe symptoms. A cut-off of ≥11 was used to identify cases of both anxiety and depression.

#### 2.2.4. Functional Mobility

The Timed Up-and-Go test (TUG) is a simple, valid, and reliable method to assess functional mobility [28], which has already been used in elderly women [29]. It is based on everyday activities and requires standing from a chair, walking three meters, turning around, and sitting down again [30]. The time required by the subject to complete this task is recorded. Longer TUG test times indicate poorer functional mobility. It has been shown that a time <12 s in the TUG test is an indication of low risk of falling [31].

#### 2.2.5. Dynamic Balance

The 3-meter Timed Tandem Walk (3MTW) Test is a functional tool used to evaluate mobility and dynamic balance. The test takes place on a marked, 3-meter long flat surface. The patient is requested to walk this distance at a comfortable pace. The time starts with the foot on the starting line and ends when the foot crosses the finishing line. Subjects walked on a marked line, stepping heel to toe, and wearing casual footwear [32]. A time >4.5 s is generally considered a likely indicator of having reported falling [33].

#### 2.2.6. Gait Speed

The OptoGait gait analysis system (Microgate Italy, Bolzano-Bozen, Italy), a device consisting of an opto-electrical detection system, was employed for gait analysis (m/s). The transmitter bar has 96 light-emitting diodes (LEDs) that communicate in the infrared spectrum. The receiver bar, which is positioned opposite it, has the same number of LEDs. The transmitter and receiver bars of the OptoGait were installed on both sides of a treadmill. To assess gait speed, participants were asked to walk for 30 s, while gait speed information was collected by the photocell system. Two familiarization and five experimental trials were performed, with a one-minute rest interval between trials. Gait speed values <1.0 m/s are associated with several fall risk factors among community-dwelling elderly people [19].

### 2.3. Sample Size Calculation

For sample size calculation at least 20 subjects per event were required in the multivariate lineal regression model [34]. Thirteen possible events were used: 7 PSQI domains, its total score, depression and anxiety scores, as well as age, BMI, and waist circumference. Therefore 260 participants were required for this analysis. The final number of participants was 271.

### 2.4. Data Analysis

Continuous variables were described using means and standard deviations, whereas categorical variables were described using frequencies and percentages. The Kolmogorov-Smirnov test was used to assess the normal distribution of variables. A bivariate correlation analysis was used to assess the possible individual ways that sleep quality, anxiety, and depression as independent variables, as well as other covariables such as age, BMI, and waist circumference, are associated with gait speed, functional mobility, and dynamic balance. In order to explore the independent associations between variables a multivariate linear regression model was employed, as well as a stepwise method for introducing variables into the model. Gait speed, functional mobility, and dynamic balance were individually introduced as dependent variables in separate models. We first looked into the bivariate correlation coefficients and variables with significant associations (*p* < 0.05) were included in the multivariate linear regression. Adjusted-R^2^ was used to calculate the effect size coefficient of multiple determination in the linear models. R^2^ can be can be considered insignificant when <0.02, small if between 0.02 and 0.15, medium if between 0.15 and 0.35, and large if >0.35 [35]. A 95% confidence level was used (*p* < 0.05). Data management and analysis were carried out using the SPSS statistical package for the social sciences for Windows (SPSS Inc., Chicago, IL, USA).

## 3. Results

Table 1 displays the descriptive data for the participants. A total of 271 women (69.18 ± 5.69 years) took part in the present study. Most participants were married or living with a partner (59.04%), had primary education or less (81.55%), and were retired (81.54%). Mean BMI was 29.96 ± 4.22 kg/ m^2^, which indicates overweight bordering on obesity, and the mean waist circumference value was 94.54 ± 9.59 cm, which indicates abdominal obesity. Scores for HADS anxiety and depression were 6.86 ± 4.21 and 5.58 ± 3.59 respectively. Regarding PSQI, sleep disturbances was the most affected domain, and its total score was 7.57 ± 4.31. Regarding the dependent variables, time values for the TUG and the 3MWT were 8.35 ± 2.08 s and 2.58 ± 0.77 s respectively, while gait speed was 1.16 ± 0.27 m/s, which indicates a low risk of falls.

The bivariate analysis (Table 2) showed that all the dependent variables considered in the present study (i.e., gait speed, TUG test, and 3MTW) significantly correlated with both HADS anxiety and depression scores, as well as with the PSQI total score. When analyzing gait speed, significant correlations were also observed with all the domains of the PSQI except for sleep disturbances and daytime dysfunction (gait speed), and sleep efficiency and sleep disturbances (functional mobility). In regard to functional mobility, lower time values in the 3MTW test were correlated with poorer sleep quality, sleep latency, and daytime dysfunction. As for the covariables included in the analysis, higher time values in the 3MTW test were associated with higher values in all three of them, while greater BMI and waist circumference were linked to longer time in completing the TUG test, and only waist circumference was negatively related to gait speed.

The multivariate linear regression analysis (Table 3) showed several independent associations with the different fall risk factors analyzed in this study. Poor sleep quality was linked to sleep efficiency, and the use of use of sleeping medication was associated with decreased gait speed (R^2^ = 0.072). Increased age, depression symptoms, waist circumference, and the use of sleeping medication were related to poorer functional mobility (R^2^ = 0.159). Lastly, the multivariate analysis revealed associations between worse dynamic balance and higher levels of anxiety and depression, and age (R^2^ = 0.127).

## 4. Discussion

The main purpose of this study was to explore the associations of sleep quality, anxiety, and depression with fall risk factors. Our results showed that elderly women are at risk of falling when they have poor sleep quality, specifically when they have a poor sleep efficiency and are administered high doses of sleep medication, as this is associated with decreases in gait speed. Furthermore, when poor sleep efficiency and high levels of depression are present, an increased risk of falling has been observed due to reduced functional mobility. Finally, when anxiety and depression are present, the risk of falling increases, since they have been inversely associated with dynamic balance. This implies that psychological factors may play an important role in predicting the risk of falls among elderly women. These results contribute to the existing knowledge of the importance of sleep quality, anxiety, and depression as fall risk factors in an elderly women population [36,37,38]. 

Recently, interest has grown regarding the identification of fall risks among the elderly, particularly concerning physical factors and their potential influence on such risks. Poor TUG test performance, slower speed gait, and poor dynamic balance have been linked to being at risk of falling [19,31,32]. In addition, it has been suggested that non-restorative sleep can, in the long term, impair motor control and cognitive functions in adult and elderly populations [39]. Besides, psychological distress such as anxiety or depression have been linked to increased difficulty in performing both cognitive and motor coordination tasks [40]. 

Our results show that gait speed, TUG test results, and dynamic balance have a significant correlation with PSQI scores, which illustrates the link between sleep quality and motor function. For gait speed we found a correlation with sleep quality measured through PSQI, which shows that elderly women with poorer sleep quality have reduced gait speed. This finding is in agreement with those of Kurose et al. [41], who obtained similar results for 102 elderly participants with cardiovascular disease concerning sleep quality and gait speed. In addition, we found a correlation between gait speed and sleep quality, where elderly women with poorer sleep quality needed more time to complete the task. This is in accordance with Del Brutto et al. [42] who showed that higher scores in sleep quality (measured by PSQI) were significantly associated with frailty in a community-dwelling population. In contrast to our findings, a study from the year 2017 showed that, after adjusting for sex in 898 elderly individuals (506 women), longer sleep duration was associated with a decrease in gait speed [43]. Finally, a relation between sleep quality and dynamic balance was found, meaning that among subjects in our sample poor sleep quality is linked to dynamic imbalance. This association between sleep quality and dynamic balance could be explained by the fact that a restorative sleep has been proved to be necessary to learn, acquire, and maintain motor behaviors [44]. Thus, people under non-restorative sleep have difficulties in learning new motor skills and even show a tendency to worsen their performance in that regard. This trend has been observed to be stronger among the elderly. Additionally, non-restorative sleep has been shown to impair motor control, which could lead to increased coordination difficulties, affecting the ability to walk and turn around fluently. Despite the relationship found in our study between sleep quality and functional mobility, it seems that other factors, such as the amount of sleep time, may play a significant role, as suggested by a study carried out in an elderly population [45]. Therefore, sleeping longer or shorter than 7–8 h has been correlated with an increase in the time required to complete the TUG test [43]. This could be explained by a decrease in the time that elderly individuals are active, which may then contribute to lower skeletal muscle mass levels and consequently to a decrease in muscle strength [46].

Our results showed that gait speed, TUG test scores, and dynamic balance have a significant correlation with both HADS scores. This highlights the need to also evaluate psychological factors such as anxiety and depression and not just functional factors in order to obtain a correct assessment of the risk of falling among the elderly. This is in agreement with recent studies [47,48] which have also reported significant positive associations between TUG test scores and depression and anxiety. While the reason for psychological distress impairing functional mobility may lie in the decreased ability to focus on the task at hand, it has also been suggested that those who are afraid of falling may experience anxiety [47], which could lead to a vicious circle of depression and self-imposed restrictions in the performance of daily activities.

In addition to functional and psychological factors, our results showed that physical factors such as age, BMI, and waist circumference are related to the risk of falling. Therefore, having a larger waist circumference negatively affects gait speed, and having higher values on any of these three physical factors negatively affects dynamic balance. Others researchers have obtained similar results [49,50]. A possible explanation for these results may be linked to an increase in sarcopenia or sarcopenic obesity, which has been associated with older age and obesity, and has been shown to have a negative impact on the risk of falls [51]. 

Finally, the multivariate linear regression analysis showed associations for gait speed, functional mobility, and dynamic balance. Gait speed was found to be negatively affected by poor sleep quality as it relates to sleep efficiency and the use of sleeping medication. Functional mobility was found to be negatively affected by older age, depression symptoms, increased waist circumference, and the use of sleeping medication. Dynamic balance was found to be negatively affected by older age and having higher levels of anxiety or depression. Taken together, when evaluating risks or when planning a treatment aimed at reducing fall risks in a population of elderly women, our scope should not be limited to physical function but include psychological and physical aspects as well given that our analyses show them to be closely linked to the risk of falls.

This study has some limitations that must be acknowledged. Its cross-sectional design does not allow for determining causal links between sleep quality, anxiety, and depression and the fall risk factors we have analyzed, and therefore any result should be interpreted with caution. In addition, although the PSQI is a validated and widely-used questionnaire to assess sleep quality, we did not employ objective measurement tools such as polysomnography or accelerometery. 

## 5. Conclusions

In conclusion, the findings of the present study suggest that, among community-dwelling postmenopausal women aged ≥60 years, sleep quality, anxiety, and depression were independently associated with three important fall risk factors such as gait speed, functional mobility, and dynamic balance. Particularly, our results showed an association of poor sleep efficiency and the use of sleeping medication with decreased gait speed; of worse functional mobility with depression and the use of sleeping medication; and lastly, of greater levels of anxiety and depression symptoms with poor dynamic balance. These findings allow us to suggest that gathering information about sleep quality, anxiety, and depression may be of help when considering fall risks in community-dwelling postmenopausal women aged 60 years and over. Future intervention studies should probably use objective measurements in order to corroborate and expand on our findings.

## Figures and Tables

**Figure 1 ijerph-17-04043-f001:**
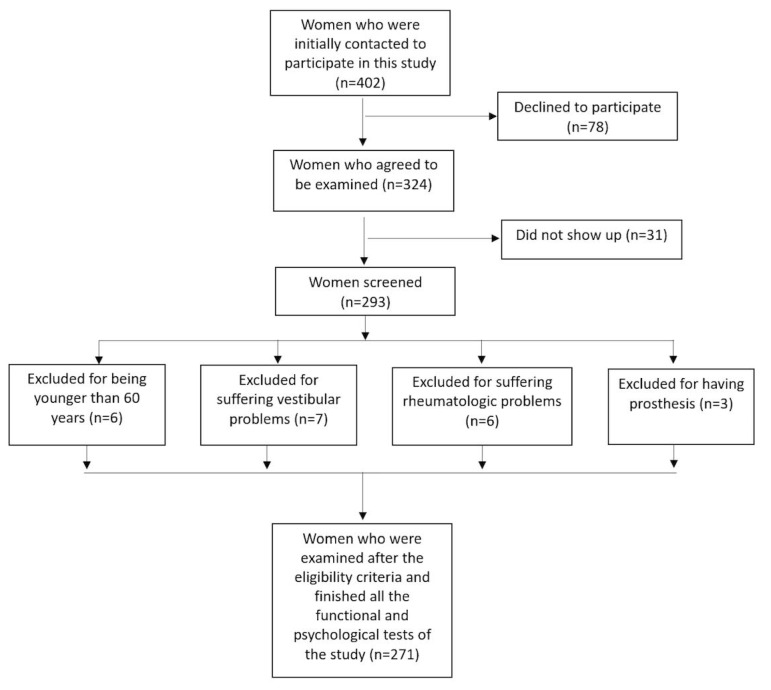
Flow chart of the study participants.

**Table 1 ijerph-17-04043-t001:** Descriptive data of the sample (*n* = 271).

Characteristics		Values	
		Mean	SD
Age (years)		69.18	5.69
Time since menopause (years)		20.41	8.34
Waist circumference (cm)		94.54	9.59
BMI (kg/m^2^)		29.96	4.22
		**Frequency**	**Percentage**
Occupational status	Retired	221	81.54
	Working	20	7.38
	Unemployed	30	11.07
Marital status	Single	3	1.11
	Married/cohabiting	160	59.04
	Separated/divorced/Widowed	108	39.85
Educational status	No formal education	83	30.63
	Primary education	138	50.92
	Secondary education	34	12.55
	University	16	5.90
Smoker	Yes	11	4.06
	No	260	95.94
		**Mean**	**SD**
HADS Anxiety score		6.86	4.21
HADS Depression score		5.58	3.59
PSQI	Sleep quality	1.14	0.83
	Sleep latency	1.38	1.02
	Sleep duration	0.87	0.92
	Sleep efficiency	0.94	1.13
	Sleep disturbances	1.56	0.72
	Use of sleeping medication	1.02	1.33
	Daytime dysfunction	0.65	0.69
	Total score	7.57	4.31
Gait speed (m/s)		1.16	0.27
TUG test (s)		8.35	2.08
3MTW (s)		2.58	0.77

BMI: Body Mass Index. HADS: Hospital Anxiety and Depression Scale. PSQI: Pittsburgh Sleep Scale Index. TUG: Timed Up and Go test. 3MTW: 3-Meter Timed Tandem Walk Test. SD: Standard Deviation.

**Table 2 ijerph-17-04043-t002:** Pearson’s correlations between variables analyzed in this study.

Variable		Gait Speed (m/s)	TUG Test (s)	3MTW (s)
PSQI	Sleep quality	−0.220 **	0.171 **	0.179 **
	Sleep latency	−0.197 **	0.162 **	0.161 **
	Sleep duration	−0.133 *	−0.020	−0.009
	Sleep efficiency	−0.250 **	0.019	0.051
	Sleep disturbances	−0.044	0.188 **	0.099
	Use of sleeping medication	−0.170 **	0.198 **	0.110
	Daytime dysfunction	−0.031	0.265 **	0.122 *
	Total score	−0.248 **	0.205 **	0.154 *
HADS Anxiety score		−0.133 *	0.318 **	0.270 **
HADS Depression score		−0.135 *	0.294 **	0.283 **
Age (years)		−0.018	0.073	0.213 **
BMI (kg/m^2^)		−0.074	0.202 **	0.178 **
Waist circumference (cm)		−0.220 **	0.171 **	0.179 **

TUG: Timed Up and Go test. 3MTW: 3-Meter Timed Tandem Walk Test. PSQI: Pittsburgh Sleep Scale Index. HADS: Hospital Anxiety and Depression Scale. BMI: Body Mass Index. * *p* < 0.05. ** *p* < 0.01.

**Table 3 ijerph-17-04043-t003:** Multivariate linear regression analyses for factors associated with gait parameters, functional mobility, and dynamic balance.

Variable		*B*	*β*	*t*	95% CI		*p*-Value
Gait speed (m/s)	Sleep efficiency	−0.05	−0.23	−3.79	−0.08	−0.03	0.000
	Use of sleeping medication	−0.03	−0.13	−2.15	−0.05	0.00	0.032
TUG test (s)	Age	0.10	0.29	5.04	0.06	0.15	0.000
	Depression	0.10	0.18	3.05	0.04	0.17	0.003
	Waist circumference	0.03	0.14	2.47	0.01	0.06	0.014
	Use of sleeping medication	0.21	0.13	2.29	0.03	0.39	0.023
3MTW (s)	Depression	0.03	0.14	1.93	0.00	0.06	0.054
	Age	0.03	0.21	3.65	0.01	0.04	0.000
	Anxiety	0.04	0.19	2.57	0.01	0.06	0.011

*B*: unstandardized coefficient. *β*: standardized coefficient. CI: confidence interval. TUG: Timed Up and Go test. 3MTW: 3-Meter Timed Tandem Walk Test.

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
