# Peer review of "Sleep Quality, Anxiety, and Depression Are Associated with Fall Risk Factors in Older Women"

_ijerph, 2020, doi:10.3390/ijerph17114043_

Round 1

Reviewer 1 Report

Minor revisions

GENERAL COMMENTS:

I have had the opportunity to review your paper and I congratulate you on the fact that your work has relevance to the IJERPH readership and addresses an important research question. It is a novel study, with significant results that address and respond to the hypothesis of the proposed study. English grammar and writing are correct. INTRODUCTION 1.      The introduction is solid and addresses the research problem, the aims are clear, and the hypothesis is well supported. MATERIALS AND METHODS 1.      In “Sociodemographic and anthropometric data” section could you explain what tools and model you used to control height, weight and waist circumference? It would be important to add it to the manuscript. 2.      Page 3, line 82. Please replace m2 by m2. Check it throughout the manuscript.

3.      Page 4, line 119-120. It’s described “"Treadmill Walking 3 119 mph or 5 km/h" is used but then you explain that the test is done at “constant speed of 3.5 km/h”. Why speed is measured if you are doing the test at constant speed? If speed is measured is because speed is self selected both on the ground or on the treadmill.4.      The independent and dependent variables are not clear, it would be important to add a new section after the "Study Participants" (“outcomes”) and explain in brief lines which are the dependent and independent variables to analyze in the study.

5.      Gait analysis. Include units (m/s)

6.      Data analysis. Please replace R2 by R2. Check it throughout the manuscript.     

RESULTS 1.      In the "Gait Speed" section, you explain that constant speed was 3.5 km/h, and then in Page 5, lines 154-155, in “Results” section show a gait speed results was 1.16 (m/s). Then, I consider that the test has been carried out on the ground at free speed or it has been carried out on a treadmill with self-selected speed. Can you answer this question and modify it in the manuscript? 2.      Page 6, line 162- 163. “r ≥ 0.xxx”, “p < x.xxx”. Please correct it.  

DISCUSSION AND CONCLUSIONS 1.      The discussion and conclusions show significant data to conclude your hypothesis, considering justified and supported conclusions. 2.      Justa an appreciation in page 8, line 266. In this line there is a comma (,) between the words “and” and “lastly”. It should be removed from the manuscript.

REFERENCES 1.      References 1-15: the years are not in bold. This must be corrected in the manuscript. 2.      Reference number 32. The title is written in uppercase letter. Please correct it and check all the manuscript. 

Reviewer 2 Report

Well written, well designed, and discussed study. Comments below are provided to make the writing less ambiguous.

Figure 1 Instead of "Did not meet Criteria..." that makes it sound like the excluding condition was the criteria, be more direct and say "Excluded because..."

Line 96 type replace "and" with "an"

Line 97 replace "other 7" with "the remaining other 7 items."

Table 1 the values are not defined, need to indicate whether the second column is Mean, standard deviation, percentage.

Line 189 re-write sentence since the relationship is not "or". Sleep issues and medications are necessarily connected. 

Line 213 "In contrast..." this study could be that they looked at people with cognitive issues. Prolonged sleep duration may be a marker of early neurodegeneration (Westwood et al, 2017). Although the relationship might be a U shape with less and more sleep indicative of cognitive issues (Chen et al, 2016).

References:

Westwood, A. J., Beiser, A., Jain, N., Himali, J. J., DeCarli, C., Auerbach, S. H., ... & Seshadri, S. (2017). Prolonged sleep duration as a marker of early neurodegeneration predicting incident dementia. Neurology88(12), 1172-1179.

Chen, J. C., Espeland, M. A., Brunner, R. L., Lovato, L. C., Wallace, R. B., Leng, X., ... & Manson, J. E. (2016). Sleep duration, cognitive decline, and dementia risk in older women. Alzheimer's & Dementia12(1), 21-33.
